# Protein Kinase A Catalytic and Regulatory Subunits Interact Differently in Various Areas of Mouse Brain

**DOI:** 10.3390/ijms21093051

**Published:** 2020-04-26

**Authors:** Carla Mucignat-Caretta, Antonio Caretta

**Affiliations:** 1Department of Molecular Medicine, University of Padova, 35131 Padova, Italy; 2Biostructures and Biosystems National Institute, 00136 Roma, Italy; 3Department of Food and Drug, University of Parma, 43100 Parma, Italy

**Keywords:** brain, cAMP, cAMP-dependent protein kinase, hippocampus, cortex, catalytic subunit

## Abstract

Protein kinase A (PKA) are tetramers of two catalytic and two regulatory subunits, docked at precise intracellular sites to provide localized phosphorylating activity, triggered by cAMP binding to regulatory subunits and subsequent dissociation of catalytic subunits. It is unclear whether in the brain PKA dissociated subunits may also be found. PKA catalytic subunit was examined in various mouse brain areas using immunofluorescence, equilibrium binding and western blot, to reveal its location in comparison to regulatory subunits type RI and RII. In the cerebral cortex, catalytic subunits colocalized with clusters of RI, yet not all RI clusters were bound to catalytic subunits. In stria terminalis, catalytic subunits were in proximity to RI but separated from them. Catalytic subunits clusters were also present in the corpus striatum, where RII clusters were detected, whereas RI clusters were absent. Upon cAMP addition, the distribution of regulatory subunits did not change, while catalytic subunits were completely released from regulatory subunits. Unpredictably, catalytic subunits were not solubilized; instead, they re-targeted to other binding sites within the tissue, suggesting local macromolecular reorganization. Hence, the interactions between catalytic and regulatory subunits of protein kinase A consistently vary in different brain areas, supporting the idea of multiple interaction patterns.

## 1. Introduction

In vivo, intracellular molecules are precisely segregated in specific compartments, to accomplish specific functions at targeted locations. An accurate location is crucial for components of the cAMP cascade [1], a small soluble second messenger that may affect many different and sometimes conflicting cellular functions. One type of cAMP effectors, the cAMP-dependent protein kinases (protein kinase A, PKA) can be considered the prototype of serine-threonine kinases, that phosphorylate a variety of proteins at different intracellular sites [2]. PKA are specifically regulated dynamic multimolecular complexes, consisting of two regulatory subunits that reversibly bind two catalytic subunits, hence inhibiting their phosphorylating activity. Upon binding two cAMP molecules to each regulatory subunit, catalytic subunits dissociate from regulatory subunits, become enzymatically active and can phosphorylate target proteins [3]. Different non-redundant isoforms of regulatory, i.e., inhibitory, subunits have been described: RIalpha (RIA), RIbeta (RIB), RIIalpha (RIIA) and RIIbeta (RIIB). They display distinctive biochemical characteristics, including different binding affinities for cAMP and for catalytic subunits, or diverse ability to bind various intracellular anchoring proteins [4]. Our laboratory showed that PKA regulatory subunits are differently localized in vertebrate brain. RIA and RIB insoluble clusters are present only in neural cells of some brain areas and appear with a specific developmental time course [5,6,7,8], while RII clusters are widely distributed and are present also on non-neural cells, for example, glial and ependymal cells, from earlier stages of development [9]. A similar distribution is detected in homolog areas of different species, such as chicken, lizard and turtles [10,11]. PKA regulatory subunits also vary in different human diseases or animal models of disease, including depression, different brain tumors and Parkinson’s disease [12,13,14,15,16,17]. 

Similar to other signaling proteins, alterations of the cAMP cascade are associated with many diseases, from diabetes and cardiovascular disorders to various cancers [18,19]. For example, mutation of the gene for the RIA regulatory subunit is present in patients with Carney complex [20], while mutations of the gene encoding PKA catalytic subunits are present in patients harboring cortisol producing adrenal tumors [21,22,23,24,25]. In vitro, both RI and RII dimers bind catalytic subunits, forming an inactive tetrameric enzyme [1], but it is still unclear whether in the brain all tethered PKA regulatory subunits bind to the catalytic subunit. This is a puzzling question, since the living organisms may present more complex, unpredicted interaction modes among PKA subunits, compared to in vitro models [26,27,28]. The aim of the present work is to examine the distribution of the PKA catalytic subunit in relation to regulatory clusters that were detected in the adult mouse brain [5,6,7,8,9]. We focused on mouse brain areas that previously showed the most prominent differences in the distribution pattern of PKA regulatory subunits, including the parietal cortex, hippocampus (Cornu Ammonis 1, CA1, subfield), amygdala, corpus striatum, stria terminalis and hypothalamus. We specifically targeted the docked, non-soluble pool of PKA catalytic and regulatory subunits because binding at precise sites is crucial for PKA targeting to specific downstream effectors, to achieve particular functions. In different areas of the brain, PKA catalytic and regulatory subunits were found to interact in different ways, paving the way to additional molecular phenotyping. Lastly, PKA was challenged with cAMP to explore the fate of catalytic subunits after release from regulatory subunits: upon cAMP addition, the catalytic subunit re-targeted to other binding sites, instead of being freely released.

## 2. Results

Double labeling experiments could give reliable results since antibodies gave specific signals and no autofluorescence was detected in the tissue (Appendix A). A consistent pattern of labeling was detected in different animals within the different conditions, representative images are shown in the figures. Here we present the data from the areas that show the most diverse pattern of labeling among the different proteins: cerebral parietal cortex (primary somatosensory barrel field cortex), hippocampus CA1b subfield, corpus striatum (caudate putamen), basolateral nuclei of amygdala, bed nuclei of the stria terminalis, intralaminar thalamic nuclei, lateral nuclei of hypothalamus, zona incerta.

### 2.1. In the Parietal Cortex, PKA Catalytic Subunit Colocalizes with PKA RI

Throughout the adult mouse cerebral cortex, immunolabeling showed that the PKA catalytic subunit was organized in discrete clusters. In Figure 1 and Figure 2, Supplementary Appendix A, the data collected on the primary somatosensory barrel field (S1BF) cortex are presented. At higher magnification, the labeling pattern could be better appreciated (Figure 1A–C, compared to Figure 1D–F. See also Appendix A). The PKA catalytic subunit largely overlapped with fluorescent Alexa488-cAMP (Figure 1I, 71.25 ± 8.25%), which binds to PKA RI [7] and characterizes a subset of cholinergic neurons [8]. At variance, Alexa488-cAMP had a statistically different distribution, since it colocalized with the catalytic subunit by only 56.93 ± 6.44% (chi-squared *p* < 0.05).

PKA RI and RII subunits were not diffuse in the cells; instead, they were organized in discrete clusters, clearly segregated (Figure 2), confirming previous data [7,8,9]. In the brain, RI bound fluorescently-tagged 8-derivatives of cAMP (Figure 2A,C), while RII did not (Figure 2D,F). Preferential binding of fluorescent cAMP to RI coupled to immunofluorescence allowed the simultaneous detection of both RI and RII, or RI and catalytic subunit in the same section. Apparently, in the cerebral cortex, the PKA catalytic subunit was mostly bound to the cAMP-binding regulatory RI subunit of PKA (88.24%, Figure 1A,G). On the contrary, a large fraction of RI did not bind catalytic subunits (45.93%, see Figure 1B, arrowheads and Figure 1H), compared to 11.76% catalytic immunolabeling not colocalizing with cAMP (Figure 1G), resulting in a statistically different distribution (chi-squared *p* < 0.0001). At a regional level, we confirm that RI clusters were restricted to neurons in some brain areas only, since RI was found in proximity of the neuronal specific markers NeuN (Appendix A) [29] or NeuroTrace (Appendix A), while RII distribution was more widespread. Although RI and RII sometimes were very close, apparently in the same cell (see also Figure 4D in [9]), in the cerebral cortex they were clearly separate (*p* < 0.0001). In summary, RII clusters in the cerebral cortex are mostly devoid of catalytic subunits, while catalytic subunits bind to RI.

### 2.2. The Amount of Colocalization between Catalytic and Regulatory PKA Varies in Different Brain Areas

Co-occurrence of PKA catalytic and RI subunits was not the rule in different brain areas, suggesting that the same proteins may interact differently in distinct brain areas. In the hippocampus, RI clusters were prominent in the pyramidal layer of Cornu Ammonis 1 (CA1) field. Figure 3A–C shows a typical CA1b subfield, where the PKA catalytic subunit was partly overlapping (60.97%) with fluorescent cAMP but significantly less than in the S1BF cortex (chi-square *p* < 0.0005, Figure 1G and Figure 3A–C,G,H). PKA catalytic subunit was also found on some cells in the stratum oriens, where no RI could be detected (see also [7]), since fluorescent cAMP labeling is restricted to pyramidal layer in proximity of a nicotinic acetylcholine receptor marker (Appendix A).

Strong PKA catalytic immunolabeling was also present in some areas where no fluorescent cAMP or RI labeling was present: for example, in the corpus striatum (Figure 3D–F) where only RII clusters were observed, as previously described [9]. Catalytic subunit was also detected on ependymal cells and choroid plexi, where no RI clusters could be observed (Appendix A), while in these areas RII clusters were previously described [9]. It is noteworthy that, in these areas, PKA catalytic labeling resembles in shape RII labeling.

A still different distribution pattern was present in the bed nucleus of stria terminalis (Figure 4D–F, H–L): catalytic subunit was close to RI/fluorescent-cAMP, but clearly distinct from it, while in a nearby area, the area amygdaloidea anterior, they are distributed differently (Figure 4G,H, chi-square, *p* < 0.0001) and colocalize (Figure 4A–C, G–I).

### 2.3. Variations in PKA Catalytic and Regulatory Subunit Distribution Do Not Match with Expression Pattern

The differences in PKA catalytic subunits distribution could possibly be linked to variations in the degree of expression in the different areas of the brain because very low expression could favor binding to high affinity binding sites, while higher expression could provide proteins also to lower affinity binding sites. Inspection of expression data on cerebral cortex, hippocampus, corpus striatum and olfactory brain revealed no apparent link to modifications in PKA catalytic distribution (Figure 5).

It may be noted that using a different technology (see Materials and Methods, Section 4.4 for details), two different laboratories reported a larger expression of RI subunits compared to catalytic or RII subunits in the mouse brain (Appendix A) but, as above, also in these cases, no obvious link was apparent between protein expression and localization in the different brain areas.

### 2.4. Retargeting of PKA Catalytic Subunit by cAMP

Lastly, we were interested in the effect of cAMP on catalytic subunit and found that the addition of cAMP induced a re-localization of the PKA catalytic subunit, but not of PKA regulatory subunits. In a control brain section, without previous exposure to cAMP, exact overlapping could be observed on large cholinergic neurons of substantia innominata where RI and catalytic subunits were present both as discrete dot-like structures and diffuse cytoplasmic labeling, outlining the shape of neuronal cell bodies and neuronal processes. The same happened in the lateral nuclei of hypothalamus (Figure 6A,B), while in the cerebral S1BF cortex catalytic and RI labeling were punctuated in shape and coincidental (Figure 1A–C. See also Appendix A). After incubation with cAMP, a different pattern of catalytic immunolabeling could be observed. Exposure to cAMP caused a complete change in the distribution of catalytic subunits, that is, their localization coincidental with regulatory subunit clusters was abolished and a fragmentary punctuated pattern appeared, appreciated at higher magnification (Figure 6C, Appendix A), while localization of RI regulatory subunits was not affected (Figure 6D and Appendix A). This effect is apparent throughout the brain; see, for example, Appendix A, in which the catalytic subunit is scattered, while Alexa488-cAMP labeling is apparent on neurons of the olfactory cortex. This labeling, but not catalytic labeling, disappears after 8Br-cAMP incubation (Appendix A).

These data suggest that upon exposure to cAMP, catalytic subunits were detached from their binding sites in regulatory subunit but were not completely solubilized since they were not entirely released in the soluble brain fraction. At least in part, they seemed to be still bound to other proteins in the brain, as shown by Figure 6C. Upon removal of cAMP from unfixed sections, it has never been possible, under the present conditions, to observe reconstitution of regulatory-catalytic subunits complexes, suggesting that cAMP removal is a necessary but not sufficient condition for PKA holoenzyme reorganization. After fragmentation of brain tissue upon exposure to cAMP, a considerable increase in the quantity of PKA catalytic subunit was expected in the soluble fraction. However, only a partial increase in partitioning of catalytic subunit between pellet and supernatant could be observed (+18.17% after cAMP exposure, Figure 6I), confirming that cAMP did not induce a massive release of PKA catalytic subunit in solution. Whereas, immunohistochemistry suggests that cAMP substantially delocalizes PKA catalytic subunit from its regulatory docking site. 

## 3. Discussion

Taken together, the present data confirm that, in the brain, a large fraction of PKA regulatory subunits is segregated into distinct subcellular compartments. They are docked at specific sites and not freely diffusible. At regional level, RI subunit clusters are present only in some neurons in specific brain areas (see Figure 1, Figure 2, Figure 3 and Figure 4) [6,7,8], while RII clusters are widely distributed, also on non-neural cells [9]. Although dual specificity A-kinase anchoring proteins (AKAP) exist [31], we did not detect mixed RI-RII clusters in the brain. Moreover, RI clusters were not observed in both primary and cell line cultures [13]. 

Our data show that in the mouse brain a large fraction of PKA catalytic subunit is normally bound to non-diffusible regulatory subunit clusters, the association with each regulatory subunit being regionally specific, consistent and not explained by variations in expression of the PKA catalytic subunit. In cells where only RII clusters can be detected (e.g., ependyma, striatum, see [9]), catalytic subunits can be bound only to RII clusters. In neurons where both RI and RII clusters are present, the reason for preferential binding to either RI or RII is not clear. In neurons in which both RI and RII clusters are present, the catalytic subunits are mainly bound to RI clusters, for example, in the cerebral cortex and in the large cholinergic neurons of the basal forebrain. Even in the same cell, some RI clusters are bound to catalytic subunits, while others are not (Figure 1A–C), suggesting a different functional condition of the same type of regulatory subunit in different microdomains. 

The binding specificity of RI for fluorescent cAMP suggests a different supramolecular organization of RI and RII in the brain, which hampers fluorescent cAMP binding to RII, at variance with in vitro experiments in which both PKA RI and RII subunits bound fluorescent cAMP [5,32]. The present data on normal brain may be relevant even if dissimilar from in vitro experiments, which may be a misleading simplified condition, possibly accounting for failure of promising therapies for different diseases [26,27,28]. It is noteworthy that in vitro PKA RI subunits have a substantially higher binding affinity for cAMP than RII [33,34]; hence, RI should bind cAMP more avidly and release catalytic subunits at lower cAMP concentration than RII subunits would do, making the RI-catalytic colocalization quite puzzling. It would be interesting to know whether the reported difference in binding constants of cAMP for RI (50–100 nM) and RII (200–400 nM) measured in vitro [35] are predictive of those in vivo, given that also the catalytic subunit behaves differently in vitro and in cells [36,37]. However, accurate measurements on living tissues are still lacking.

Surprisingly, in some brain areas like the cerebral cortex, we found the catalytic subunit bound to RI and not to RII; this is an unexpected finding, given the higher binding affinity of RI for cAMP. This result is possibly related to subcellular conditions and selective compartmentalization of cAMP and cAMP transduction machinery. Anyhow, the colocalization of RI and catalytic subunit is not always the case; in the hippocampal cornu Ammonis subfield 1 (CA1) pyramidal layer, RI and catalytic subunits are only partially coincidental, while in the bed nuclei of the stria terminalis (Figure 4D–F) catalytic subunit clusters are in close proximity but clearly distinct from RI, being likely bound to RII clusters. All these different conditions strongly suggest variations in biochemical properties of the same proteins, when located in the complex organ. The relative localization reported here does not match those of any known protein kinase A anchoring protein, or associated protein like protein phosphatases or phosphodiesterases [1]. It may be noted that no data are available on regional localization within the entire brain. Our data point to a complexity in PKA regulatory/catalytic interactions, which could not be otherwise predicted.

In the brain, upon cAMP binding to regulatory subunits, catalytic subunits were completely released from their regulatory subunit binding sites but did not freely diffuse (Figure 6 and Appendix A); they switch to other (presumably lower affinity) binding sites, that are on insoluble fraction, where they can possibly perform their phosphorylating activity, as recently described in cell cultures [38]. Partitioning between pellet and supernatant suggests that upon addition of cAMP, only a minor fraction of catalytic subunits is actually solubilized, despite being released by regulatory subunits. Since regulatory subunits are in a molar excess compared to catalytic subunits, we cannot rule out the recapture by regulatory binding sites, but we may rule out the recapture by the same type of PKA regulatory subunit since it is apparent that the catalytic distribution after being released was different from the regulatory distribution, which did not change. It remains open the possibility, described in embryonic kidney cells, of catalytic myristoylation, which could target them to the plasma membrane when released from RII [39], possibly linking this process to lipid rafts [40]. Lastly, data obtained from protein quantification gives a different picture from gene expression, since regulatory subunits are in molar excess compared to catalytic subunit in brain protein extracts [39], while data obtained from RNA, with different techniques and in different laboratories, consistently give the opposite picture (Figure 5 and Appendix A); an additional warning on the complexity of biological systems.

The functional meaning of interactions between regulatory clusters and catalytic subunits is challenging. According to the commonly held view, catalytic subunit phosphorylating activity is allowed only after dissociation from regulatory subunits, as a consequence of cAMP binding. The clusters of regulatory and catalytic subunits may fulfill different functions in cells. They may slow down or hinder cAMP diffusion, given the high local concentration of cAMP binding sites. They may concentrate catalytic subunits close to target proteins. Alternatively, under local conditions of low cAMP concentration, they may act as inhibitory traps for catalytic subunits, thus creating phosphorylation-free intracellular microdomains. 

However, the standard model of cAMP protein kinase functioning has been recently questioned and is becoming more intricated. In vitro, depending on concentrations of regulatory and catalytic subunits and in the presence of saturating concentrations of cAMP, the possibility of ternary complexes has been proposed, that is cAMP-regulatory subunits type I-catalytic subunits assembled together. Under these conditions, the catalytic subunit remains inactive [41]. A still different possibility has been suggested in cell culture [42]; the catalytic subunits can be enzymatically active when still bound to RII regulatory subunits. Furthermore, binding of cAMP to PKA is regulated by phosphorylation of the RII subunit, which also modulates the time course of its action [43]. Lastly, the relationship of PKA holoenzyme with different phosphodiesterases may drive specific downstream effects [44]. Noticeably, a direct phosphodiesterase/RIalpha interaction has been already described [45].

The changing concentrations of regulatory and catalytic subunits, assembly–disassembly of macromolecular complexes because of oscillating intracellular concentration of second messengers, the multiple isoforms and multiple intracellular localizations create a huge number of possible combinations and make it hard to predict what actually happens inside a neuron in the brain. It may be possible that the same system upon changing local conditions may perform different tasks even within the same intracellular microdomain.

## 4. Materials and Methods 

### 4.1. Animals

Experiments were approved by the competent authority (OPBA, University of Padova, and Italian Ministry of Health, N. 43F3ENEYD, approved 22 February 2018) and were performed according to European laws on animal experiments (directive 2010/63/EU). All efforts were made to minimize the number of animals used and their suffering. Twenty male and female CD-1 mice 35–45 days old were reared on 12:12 h light cycle (light on at 6:00), with food and water ad libitum, at 23 ± 1 °C. They were anesthetized with halothane and sacrificed by cervical dislocation. The brains were removed and immediately frozen in nitrogen-cooled pentane. 

### 4.2. Immunofluorescence

The procedure has already been described in detail [8,46]. Experiments were replicated at least three times, and representative micrographs are presented in the figures. In order to provide precise localization of brain areas, whole brains were cut on cryostat in coronal or horizontal sections (16 µm), put on a coverslip and air-dried, washed in 100 mL phosphate-buffered saline (PBS, 150 mM NaCl, 10 mM phosphate buffer, ethylene-diaminetetraacetic acid 0.5 mM, pH 7.4) for 30 min at 25 °C and then fixed in formalin (10% in PBS) for 1 h, rinsed in PBS-Triton 2% for 30 min and air-dried. Every fifth slide was subsequently processed for immunofluorescence, followed by Nissl staining to provide identification of brain areas [47]. For immunofluorescence, after blocking non-specific binding (30 min with 0.4% bovine serum albumin in PBS), alternate sections were incubated with rabbit anti-RI, RII, Catalytic subunit (respectively: sc-907, sc-909, sc-903, Santa Cruz Biotechnology, Santa Cruz, CA, USA) or mouse anti-NeuN (Chemicon mAb-377) antibodies 1:200 in PBS, at room temperature overnight. After extensive washing in PBS, sections were incubated at 37 °C for 30 min with either Alexa 488- or Alexa 594-conjugated anti-rabbit or anti-mouse secondary antibody (Molecular Probes, Eugene, OR, USA), 1:250 in PBS. Alternatively, NeuroTrace 530/615 (ThermoFisher, Monza, Italy) was used to visualize neurons, α-bungarotoxin labeled with fluorescein isothiocyanate (Sigma, Milan, Italy) was used to label α7-nicotinic acetylcholine receptors, and DAPI (Sigma, Milan, Italy) was used for nuclei.

For colocalization experiments, selected sections were mounted in PBS containing fluorescent nucleotides: 100 nM 8-(5-thioacetamidofluorescein)-adenosine 3′,5′-cyclic monophosphate (SAF-cAMP) or 8-(5-thioacetamidotetramethylrhodamine)-adenosine 3′,5′-cyclic monophosphate (SAR-cAMP), or 250 nM 8-(2-fluoresceinylthioureidoaminoethylthio)-adenosine 3′,5′ -cyclic monophosphate (8-Fluo-cAMP), or 200 nM 8-(Alexa488)- adenosine 3′,5′-cyclic monophosphate (Alexa488-cAMP), as previously described [5,32]. SAR-cAMP and 8-Fluo-cAMP were tested at least on three different mice, SAF-cAMP and Alexa488-cAMP were tested on at least 6 mice. SAF-cAMP and SAR-cAMP were synthesized [46,48], Alexa488-cAMP was from Molecular Probes (Eugene, OR), 8-Fluo-cAMP was obtained from BioLog (Bremen, Germany). Although SAR-cAMP, SAF-cAMP and 8-Fluo-cAMP can bind in vitro also to the RII soluble type [32,33], they visualize only RI clusters on histological brain sections, due to specific binding to RI under equilibrium conditions [5,6,46]. They are readily displaced by 50 µM 8-Br-cAMP, resulting in specific abolition of the fluorescent cAMP labeling [5].

In order to evaluate the effect of cAMP on catalytic subunit immunolabeling, adjacent sections were examined. Before formalin fixation, to avoid protein–protein crosslinking [49], sections were either exposed for 2, 10 or 30 min to PBS added with 50 µM cAMP and 500 µM 3-isobutyl-1-methylxanthine (IBMX), a noncompetitive phosphodiesterase inhibitor, or to PBS without cAMP (control), then rinsed in PBS, fixed with 10% formalin for 1 h and processed for immunofluorescence.

The following controls were run: pre-absorption with immunogenic peptide, omission of the primary antibody, incubation with rabbit pre-immune serum, omission of the secondary antibody, photographs with both red and green filters to highlight possible autofluorescence in the tissue due to lipofuscins, incubation with excess cAMP and 8Br-cAMP to displace fluorescent 8-derivatives of cAMP, pre-incubation of RI or RII antibodies with excess soluble PKA RI or RII purified as previously described to abolish RI and RII antibody binding to antigens (for details on the control conditions, see: 5, 7–11, 46). Negative controls were run in each experiment and resulted unlabeled. Positive controls were sections of tissue (different human brain tumors, different mouse organs, and tissue from different species) and cell cultures, which are known to express the antigen: these were run in each experiment. No difference was apparent between males and females in labeling of sexually dimorphic areas (accessory olfactory bulb, hypothalamic medial preoptic area); hence, the data from both sexes were pooled.

### 4.3. Western Blot

All chemicals were from Sigma, Milan, Italy, unless otherwise stated. Partitioning of catalytic subunits has been tested in triplicate between soluble and insoluble (pelletable) brain fraction, upon exposure to cAMP. The cerebral cortex (20 mg) was rapidly dissected and homogenized in 40x volume (800 µL) of PBS (140 mM NaCl, phosphate buffer 10 mM pH = 7.4, ethylenediaminetetraacetic acid 1 mM), 2 mM phenylmethanesulfonylfluoride, protease inhibitor cocktail (Roche, Mannheim, Germany) and 10 mg/mL soy-bean trypsin inhibitor. Detergents were not added because on unfixed tissue they modify the PKA catalytic distribution and partitioning, by solubilizing it (unpublished observations on biochemical preparations from rodent and chicken brain homogenates, confirmed by immunofluorescence on brain sections, in which PKA catalytic subunit cannot be detected if detergent treatment is done before formalin fixation. See Appendix A). The original sample was divided in two: one sample was an untreated control, the second sample was incubated for 2 min with 50 µM cAMP and 500 µM 3-isobutyl-1-methylxanthine (IBMX) in the same homogenization buffer solution with protease inhibitors as above. Samples were centrifuged at 10,000 g for 10 min. The supernatants were carefully removed and the pellets were resuspended in 2000 µL PBS. The pellet was diluted 5× in comparison to the supernatant since most proteins are in the pellet. After spectrophotometric protein quantification (Bradford assay, BioRad, Richmond, CA, USA), five microliters of the properly diluted supernatant and pellet were loaded on 12% sodium dodecyl-sulfate (SDS) polyacrylamide gel (BioRad, Richmond, CA, USA), blotted onto nitrocellulose membrane (Whatman International, Maidstone, England), blocked in 3% gelatin in Tris buffer 20 mM, NaCl 500 mM, pH 7.5 and incubated overnight with anti-PKA catalytic subunit (sc-903, Santa Cruz Biotechnology, Santa Cruz, CA, USA, 1:5000). The secondary antibody conjugated with horseradish peroxidase (Sigma, 1:7500) was incubated for four hours and developed with chemiluminescence (Advanced ECL, Amersham, Milan, Italy). To evaluate proper transfer, the gel was also stained with Coomassie after transfer. The use of different housekeeping proteins has been questioned for inter-subject variability or possible sensitivity to experimental conditions [50,51,52]; hence, the cumulative intensity of the respective lane, stained with Sudan black, was used as loading control [30], the background was subtracted and the value of the band normalized to the loading control. Antibody specificity was tested in Western blots and immunohistochemistry by preincubating the primary antibody with the blocking peptide. 

### 4.4. Gene Expression Data

Data on the expression of PKA catalytic subunit on the most relevant areas described here (cerebral cortex, hippocampus, corpus striatum and olfactory brain, in the original FANTOM5 database nomenclature) were accessed at the EMBL-EBI public repository on May 2019 (https://www.ebi.ac.uk/gxa/genes/ENSMUSG00000005469?bs=%7B%22mus%20musculus%22%3A%5B%22ORGANISM_PART%22%5D%7D&ds=%7B%22kingdom%22%3A%5B%22animals%22%5D%7D#baseline). Data and genomic tools are described in http://fantom.gsc.riken.jp/5/.

Data on PKA expression in the mouse brain obtained using a different technology (Affymetrix microarrays) by two different laboratories [53,54] were explored with BioGPS: www.bioGPS.org [55,56,57] and BrainStars (www.brainstars.org), respectively; heatmaps were generated with Morpheus (https://software.broadinstitute.org/morpheus/). 

### 4.5. Analysis

Western blots were acquired linearly with an Epson scanner at 1200 dpi and converted to 16-bit gray scale; only non-saturating bands were considered for intensity measurements. For fluorescence microscopy, digital RGB photographs were captured on Leica DMR microscope (objectives 20×, 40×, and 100× oil immersion, numerical aperture 1.30) at 768 × 582 pixels, using the same parameters within each experiment. Double labeled figures were obtained by merging the two original files and adjusting the brightness by 0%–10% for presentation. Imaging experiments were repeated three times; within each brain, three (for smaller areas) to six sections were examined for each area. Quantification was carried out on single-channel images extracted from the originals, using the software ImageJ to count maxima, defined as pixels of which the intensity is 10 or 20 units above the nearest. Colocalization was evaluated by superimposing two different channels and counting overlapping and non-overlapping maxima for both channels. For each image, a mean of 1205.18 ± 122.85 (range: 633–1951) maxima were counted. Chi-squared test was used to compare the distribution between different areas or between different channels, *t*-test was used to compare the number of single and double-labeled points. All images were handled using GraphicConverter 9, statistical analyses were carried out with GraphPad Prism 5.

## 5. Conclusions

Many questions ensue from these data. Why catalytic subunit binds to RI in some areas, while it should be easily released? Why in the same cortical neurons only some RI clusters are bound to catalytic subunits while others are not? Why in other brain areas are catalytic subunits preferentially bound to RII clusters? After being released from regulatory subunits, to which proteins do catalytic subunits bind? The factors responsible for these effects in the brain are still unknown and need to be investigated. In conclusion, the present data highlight the variety of conditions in which PKA regulatory and catalytic subunits may normally interact in different areas of the brain, which are not directly related to their in vitro properties.

## Figures and Tables

**Figure 1 ijms-21-03051-f001:**
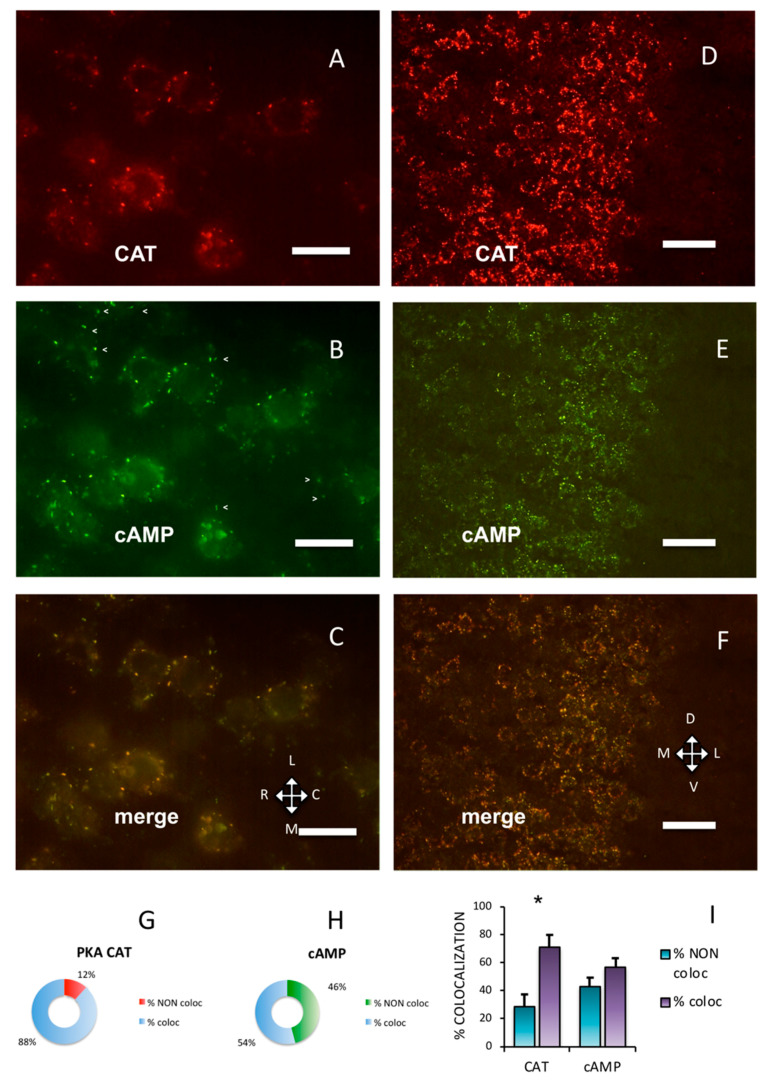
Protein kinase A (PKA) catalytic subunit colocalizes with cAMP in the cerebral parietal cortex. (**A**) Catalytic subunit immunolabeling (CAT) in the S1BF cortex, pia on the top. (**B**) Fluorescent Alexa488-cAMP (cAMP) in the same field. Arrowheads mark some cAMP-binding clusters in which no catalytic subunit is apparent (see Figure 1A,C). (**C**) Merge of A and B, showing superimposition (yellow). **A**–**C**: Horizontal section. L: lateral, M: medial, C: caudal, R: rostral. (**D**) Catalytic subunit immunolabeling at a lower magnification in S1BF cortex. Pia on the right. (**E**) Same field, fluorescent Alexa488-cAMP. (**F**) Merge of **D** and **E**, showing superimposition of the two signals. **D**–**F**: Coronal section. D: dorsal, V: ventral. Scale bar, 10 µm (**A**–**C**), 25 µm (**D**–**F**). G,H: quantification of superimposition in C (*n* = 806). (**G**) Percentage of PKA catalytic immunolabeling colocalizing (% coloc, light blue, *n* = 255) or not (% NON coloc, red, *n* = 30) with fluorescent cAMP in C. (**H**) Percentage of fluorescent cAMP colocalizing (% coloc, light blue, *n* = 357) or not (% NON coloc, green, *n* = 164) with PKA catalytic immunolabeling in C. (**I**) Percentage of colocalization (coloc, violet) and non-colocalization (NON coloc, blue) of catalytic immunolabeling (CAT) and fluorescent Alexa488-cAMP (cAMP) in three different experiments (*n* = 3389); the number of colocalizing points is significantly higher than non-colocalizing for catalytic subunit (*, 1020 vs. 493, *t*-test, *p* = 0.015), while it is not different for fluorescent cAMP (colocalizing 1115 vs. 762 non-colocalizing *t*-test, *p* = 0.467). Mean + SEM are shown.

**Figure 2 ijms-21-03051-f002:**
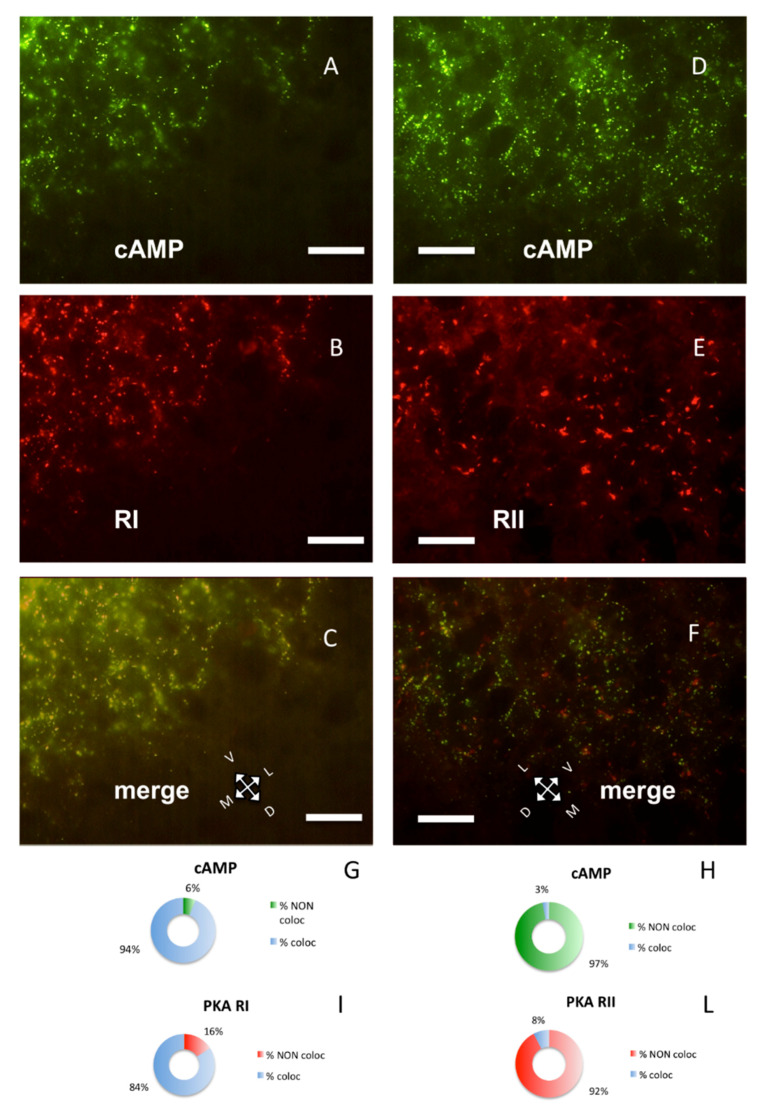
Parietal cortex coronal sections, scale bar: 10 µm. (**A**) Alexa488-cAMP (green) labeling of the cerebral S1BF cortex, pia on the lower right. (**B**) In the same field, RI immunolabeling (red). (**C**) Merge of **A** and **B**, showing coincidence of fluorescent cAMP and RI (yellow). (**D**) Alexa488-cAMP labeling (green) of the cerebral S1BF cortex, pia on the lower side. (**E**) Same field, RII immunolabeling (red). (**F**) Merge of **D** and **E** shows no colocalization of red and green signals. **G**–**I**: Quantification of superimposition in **C** (*n* = 1045). (**G**) Percentage of colocalization of cAMP (% coloc, light blue, *n* = 454) or not (% NON coloc, green *n* = 30) with PKA RI in **C**. **H**–**L**: Quantification of superimposition in **F** (*n* = 1426). (**H**) Percentage of colocalization of cAMP (% coloc, light blue, *n* = 31) or not (% NON coloc, green, *n* = 987) with PKA RII in **F**. (**I**) Percentage of colocalization of PKA RI immunolabeling (% coloc, light blue, *n* = 471) or not (% NON coloc, red, *n* = 90) with cAMP signal in **C**. (**L**) Percentage of colocalization of PKA RII immunolabeling (% coloc, light blue, *n* = 31) or not (% NON coloc, red, *n* = 377) with cAMP signal in **F**.

**Figure 3 ijms-21-03051-f003:**
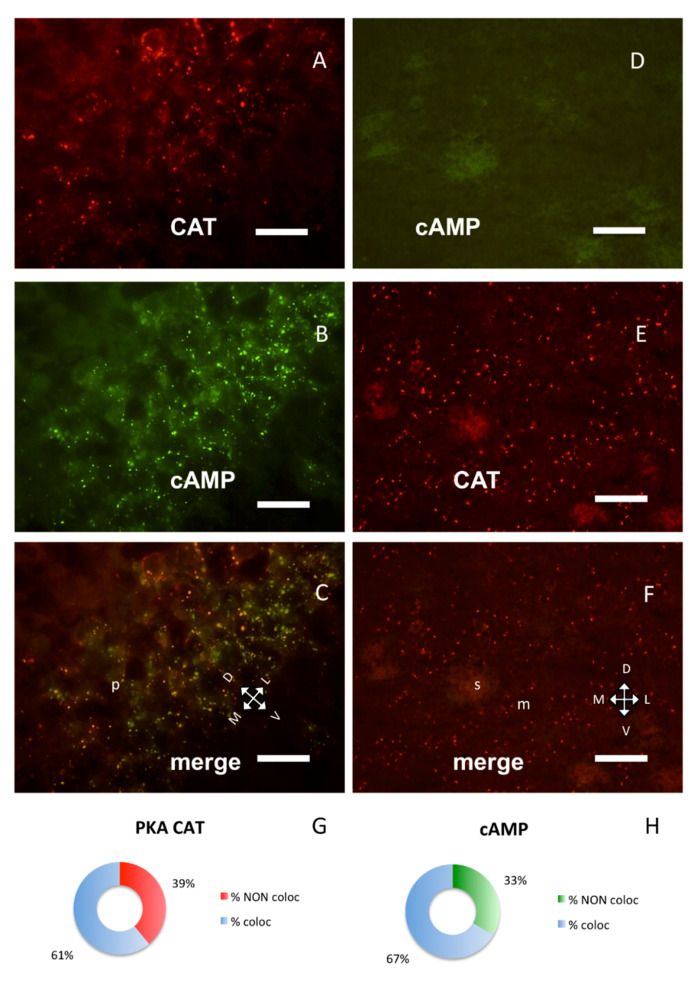
PKA catalytic subunit in hippocampus and corpus striatum, coronal sections. (**A**) Catalytic subunit immunolabeling in the pyramidal layer of CA1b subfield of hippocampus, dorsal on upper left. (**B**) Same field, Alexa488-cAMP. (**C**) Merge of **A** and **B**, showing partial colocalization of the two signals; *p*: pyramidal layer. (**D**) In the corpus striatum, no labeling can be observed with Alexa488-cAMP. (**E**) Catalytic subunit immunolabeling in the same field. (**F**) Merge of **D** and **E**. **D**–**F**: Dorsal on the top; m: matrix, s: striosomes. Scale bar, 10 µm (**A**–**C**), 25 µm (**D**–**F**). **G**,**H**: Quantification of superimposition in **C** (*n* = 1426). (**G**) Percentage of colocalization of PKA catalytic immunolabeling (% coloc, light blue, *n* = 364) or not (% NON coloc, red, *n* = 233) with cAMP in **C**. (**H**) Percentage of colocalization of cAMP (% coloc, light blue, *n* = 477) or not (% NON coloc, green, *n* = 239) with PKA catalytic immunolabeling in **C**.

**Figure 4 ijms-21-03051-f004:**
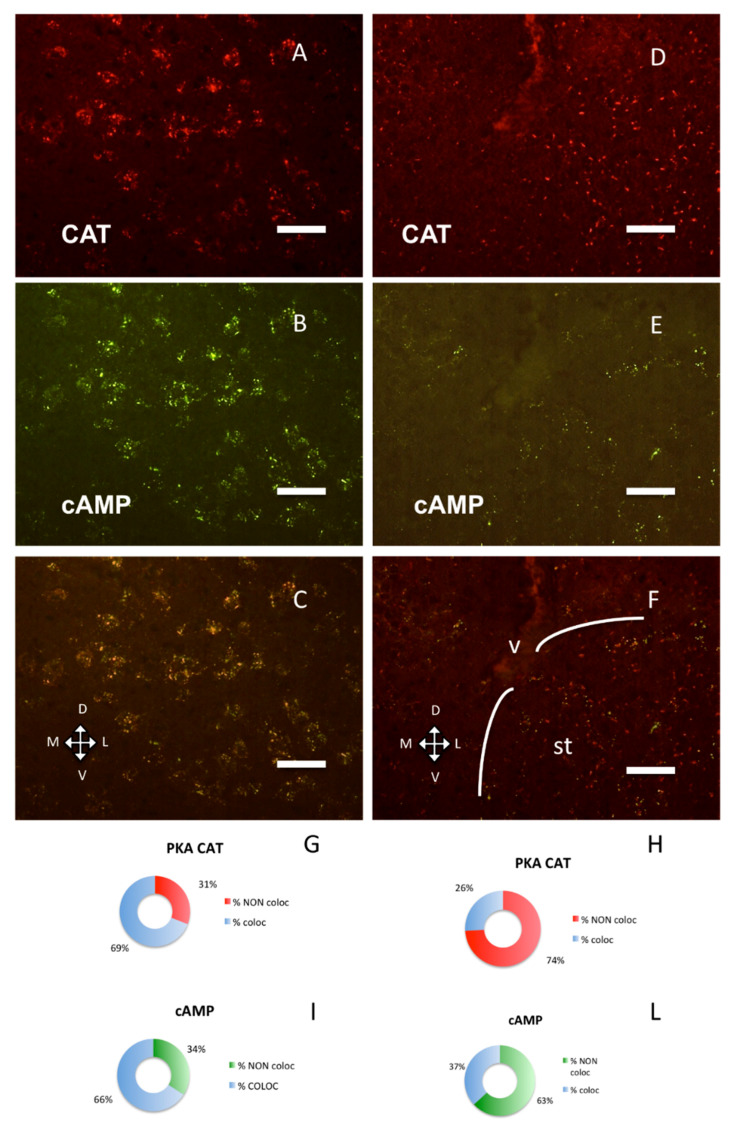
PKA catalytic subunit in the amygdala and stria terminalis, coronal sections, dorsal on the top. Scale bar, 25 µm. (**A**) Catalytic subunit immunolabeling in the basolateral nuclei of amygdala. (**B**) Same field, Alexa488-cAMP. (**C**) Merge of **A** and **B**. (**D**) Catalytic subunit immunolabeling in the bed nuclei of the stria terminalis. (**E**) Alexa488-cAMP in the same field. (**F**) Merge of **D** and **E**; v: lateral ventricle. The white lines enclose the bed nucleus of the stria terminalis (st). **G**–**I**: quantification of superimposition in **C** (*n* = 1559). (**G**) Percentage of colocalization of PKA catalytic immunolabeling (% coloc, light blue, *n* = 528) or not (% NON coloc, red, *n* = 237) with cAMP in **C**. **H**–**L**: quantification of superimposition in F (*n* = 679). (**H**) Percentage of colocalization of PKA catalytic immunolabeling (% coloc, light blue, *n* = 118) or not (% NON coloc, red, 311) with cAMP in **F**. (**I**) Percentage of colocalization of cAMP (% coloc, light blue, *n* = 523) or not (% NON coloc, green, *n* = 271) with PKA catalytic immunolabeling in **C.** (**L**) Percentage of colocalization of cAMP (% coloc, light blue, *n* = 92) or not (% NON coloc, green, *n* = 158) with PKA catalytic immunolabeling in **F**.

**Figure 5 ijms-21-03051-f005:**
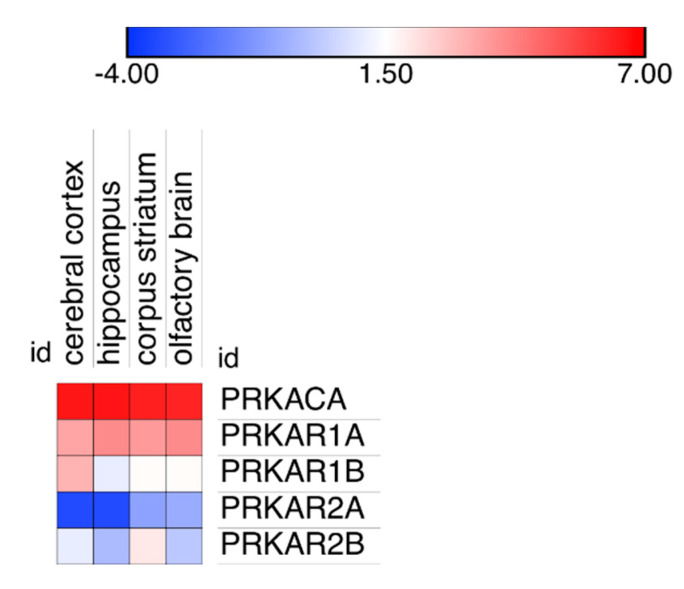
Expression level of the PKA catalytic and regulatory gene products in four different areas of the brain: cerebral cortex, hippocampus, corpus striatum and the olfactory brain. Data are plotted as Log(2) of the median of all expression data, reported in the original database as Transcripts per Million (TPM).

**Figure 6 ijms-21-03051-f006:**
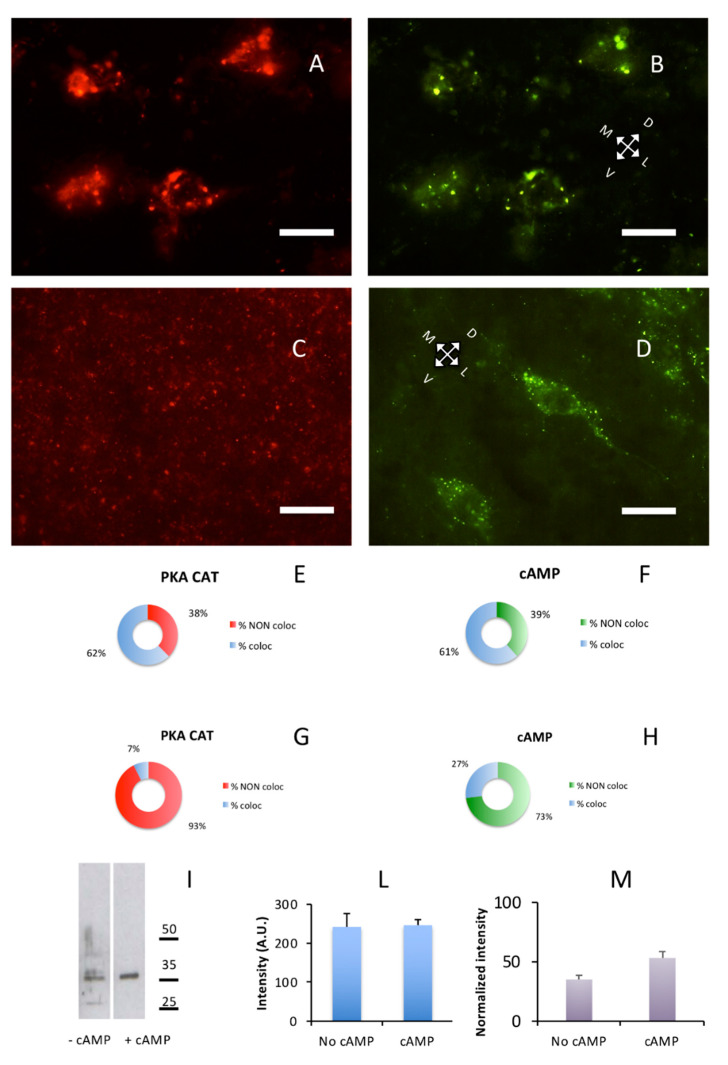
cAMP induces redistribution of PKA catalytic subunit. Coronal sections. Scale bar, 10 µm. (**A**) PKA catalytic subunit immunolabeling in the hypothalamic lateral nuclei. (**B**) Same field as A, Alexa488-cAMP. (**C**) After addition of cAMP for ten min, catalytic subunit (red) relocate as shown by immunolabeling. (**D**) Same field as C, Alexa488-cAMP. **E**,**F**: Quantification of superimposition in A and B (*n* = 633). (**E**) Percentage of colocalization of PKA catalytic immunolabeling (% coloc, light blue, *n* = 202) or not (% NON coloc, red, *n* = 122) with cAMP in A and B. (**F**) Percentage of colocalization of cAMP (% coloc, light blue, *n* = 190) or not (% NON coloc, green, *n* = 119) with PKA catalytic immunolabeling in **A** and **B**. **G**,**H**: Quantification of superimposition in **C** and **D** (*n* = 1951). (**G**) Percentage of colocalization of PKA catalytic immunolabeling (% coloc, light blue, *n* = 114) or not (% NON coloc, red, *n* = 1414) with cAMP in **C** and **D**. (**H**) Percentage of colocalization of cAMP (% coloc, light blue, *n* = 114) or not (% NON coloc, green, *n* = 309) with PKA catalytic immunolabeling in **C** and **D**. (**I**) Western blot of brain soluble fraction before (−cAMP lane) and after addition of cAMP (+cAMP lane). Only a partial effect can be appreciated, indicating that addition of cAMP does not completely solubilize the PKA catalytic subunit. Instead, if catalytic subunit is released from regulatory subunit (see also Appendix A), it is bound by other complexes, suggesting short-range local interactions. (**L**) Quantification of loading controls for lanes shown in **I**. Since an effect of cAMP treatment cannot be excluded also for housekeeping gene products, the cumulative intensity of each lane was used as a reference for normalization [30]. (**M**) Protein levels of PKA catalytic subunits (**I**) normalized to loading controls (**L**).

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
