# Peer review of "Protein Kinase A Catalytic and Regulatory Subunits Interact Differently in Various Areas of Mouse Brain"

_ijms, 2020, doi:10.3390/ijms21093051_

Round 1
Reviewer 1 Report
The authors described the distribution of PKA catalytic subunit in relation to regulatory clusters in the cerebral parietal cortex, hippocampus, corpus striatum, amygdala, and stria terminalis. In addition, they showed that cAMP could induce redistribution of PKA catalytic subunit, but not regulatory subunit, in the hypothalamus.
The study is well done and comprehensive and one issue should be addressed for consideration of publication.
- The authors should specify the number of brain slices and total cells to be analyzed in the measurement of PKA catalytic subunit, cAMP, and colocalization of these components in each brain area in the "Results" and "Materials and Methods".
Author Response
REV. 1: The authors described the distribution of PKA catalytic subunit in relation to regulatory clusters in the cerebral parietal cortex, hippocampus, corpus striatum, amygdala, and stria terminalis. In addition, they showed that cAMP could induce redistribution of PKA catalytic subunit, but not regulatory subunit, in the hypothalamus.
The study is well done and comprehensive and one issue should be addressed for consideration of publication.
AUTHORS: Thank you for your kind consideration.
- REV. 1: The authors should specify the number of brain slices and total cells to be analyzed in the measurement of PKA catalytic subunit, cAMP, and colocalization of these components in each brain area in the "Results" and "Materials and Methods".
AUTHORS: The Reviewer is right, we apologize for this omission. We have now included all the relevant numerical information in legends to figures 1-4 and 6 and supplementary S1 to S6, and in Materials and Methods (lines: 383-384 and 459-460, 464-465).
Here is the detail of the lines for figure legends: 114-122, 127-138, 169-172,181-188, 235-243.
In order to preserve readability, we have included all the numbers in the figure and supplementary figure legend and not repeated them in the Results main section, which is a descriptive summary of results.
Reviewer 2 Report
Mucignat-Caretta extensively investigated localizations of PKA-catalytic and regulatory subunits and their interactions in mouse brain. The authors have shown that interactions between catalytic and regulatory subunits vary in different brain areas and the catalytic subunits are not fully released from the regulatory subunit in response to increases of cAMP. As a next study, it is important to perform real time imaging to confirm findings in this study. The results are interesting and clear. However, I have concerns as follows.
Major concern;
1) The authors showed their observations in different brain areas. However, brain areas studies in this manuscript are still large. Therefore, the authors should describe the exact brain regions they imaged, in the figure (for example, the exact area in the CA1).
2) There are several types of cells including excitatory neurons, inter (inhibitory) neurons, glia, astrocyte in each brain area. Therefore, the authors should describe which type of neurons they imaged.
3) Additionally, it is also important to describe the localization of target molecules in the cells (nucleus, cell body, axon, spine, dendrites…) although the authors seemed to image nucleus and cell body in the cells.
4) Figure 6 is interesting but requires control experiments using 8-Br-cAMP.
Minor comment
The authors seemed to use several types of fluorescent cAMPs (SAR-cAMP, SAF-cAMP and 8-Fluo-cAMP). The authors should indicate which probe was used in each experiment.
Author Response
REV. 2: Mucignat-Caretta extensively investigated localizations of PKA-catalytic and regulatory subunits and their interactions in mouse brain. The authors have shown that interactions between catalytic and regulatory subunits vary in different brain areas and the catalytic subunits are not fully released from the regulatory subunit in response to increases of cAMP. As a next study, it is important to perform real time imaging to confirm findings in this study. The results are interesting and clear. However, I have concerns as follows.
AUTHORS: Thank you for your kind consideration, we will keep this suggestion as a priority for our next studies.
REV. 2: Major concern;
1) The authors showed their observations in different brain areas. However, brain areas studies in this manuscript are still large. Therefore, the authors should describe the exact brain regions they imaged, in the figure (for example, the exact area in the CA1).
AUTHORS: We thank the Reviewer for asking this and apologize for not having added this information in the first version of the manuscript – we were afraid of adding too much neuroanatomical details. As requested, brain areas have been more exactly identified throughout the text, in particular in the Results and in figure/supplementary legends. However, as a guide for readers less familiar with neuroanatomy, we kept more general terms (for example, ‘parietal cortex’) in some cases (e.g. in figure titles), in addition to the specific terms, which are used in the body of the figure legends and in the Results and Supplementary Material.
REV. 2: 2) There are several types of cells including excitatory neurons, inter (inhibitory) neurons, glia, astrocyte in each brain area. Therefore, the authors should describe which type of neurons they imaged.
AUTHORS: We identified the fluorescent-cAMP labelled neurons as a sub-type of cholinergic neurons by colocalization with choline acetyl-tranferase and vesicular acetylcholine transporter (Mucignat-Caretta,Molecular Brain Research, 2000, 80, 233–236), which in the adult mouse are present in olfactory, entorhinal, frontal, parietal, occipital and cingulate cortex, medial septal nuclei, vertical and horizontal limb of the diagonal band, basolateral and medial amygdala, hippocampus pyramidal layer (excluding CA2) and dentate gyrus granular layer, substantia innominata, zona incerta, medial and reticular thalamic nuclei, hypothalamus, periacqueductal gray, brainstem reticular formation, parabrachial nuclei, medial vestibular nucleus, granular layer of the vestibular cerebellum, substantia gelatinosa of Rolando in the spinal cord dorsal horn.
Hence, where PKA catalytic subunits colocalizes with fluorescent cAMP, it is located in cholinergic neurons. A statement with a reference has been added on this, at lines 85-86.
Concerning stratum oriens catalytic immunolabelled cells, they were tentatively identified as GABAergic interneurons, but since we do not present colocalization images, this information was omitted from the text. However, the focus of the present manuscript is not on the identification of single cell types, but in showing that inside the brain cAMP-dependent protein kinase subunits do behave differently in the various areas, hence each model we may use (cell cultures, for example) may be dramatically different from in vivo brain tissue. This issue is not entirely appreciated until now, since many excellent works are done on single cells and simply export the findings to the brain. We would like to make a warning statement on this by providing the present data.
Moreover, there are technical reasons for which we did not perform triple labelling in this work, in the slices in which we performed the measurements:
- secondary peaks of emission of fluorophores may interfere with excitation/emission wavelength of other fluorophores: the more fluorophores you use, the more chances you have to get artifacts due to secondary excitements. This is an issue which cannot be resolved with small slit band-pass barrier filters (we performed some trials when starting this work), by collecting light from a window spanning only 5-15 nm, because the signal would be reduced dramatically, thus missing most of the ‘hotspots’ we detected, hence losing true signals and unacceptably skewing results. This issue of course cannot be resolved by electronic means, working on acquired images. It can be faced only through the recording of technically sound images, whose intensity is reliable and signals do not cross from one channel to the other.
- Another reason to avoid band-pass filters when colleting emitted light refers to the red shift in emission wavelengths upon binding to regulatory binding sites, that was observed by us and others using fluorescent cAMP (Mucignat-Caretta and Caretta Biochim. Biophys. Acta 1997, 1357, 81; Schwede et al, Biochemistry 2000, 39, 8803). If short band-pass filters are used, it is possible to simply loose the largest signal because it may fall outside the recording wavelength window, resulting in unreliable dataset.
Hence, in order to perform robust measurements on colocalization, it was necessary to use the least number of fluorophores in the single slice, that is two (we used ‘red’ and ‘green’ fluorophores): one for fluorescent cAMP and one for catalytic subunit (or for one of the regulatory subunits, in different brain slices).
While technically feasible for imaging purposes, triple or quadruple (for example by adding also DAPI as a nuclear marker and a far-red emitting secondary antibody for a third protein) fluorescent labelling would result in significant quenching or cross over of some signals, hence are not suitable to perform measurements.
In this manuscript we preferred to provide consistent measurements to compare the different brain areas, leaving to our next works to explore the colocalization of multiple markers.
REV. 2: 3) Additionally, it is also important to describe the localization of target molecules in the cells (nucleus, cell body, axon, spine, dendrites…) although the authors seemed to image nucleus and cell body in the cells.
AUTHORS: We agree with the Reviewer and keep as of utmost importance this advice. As an aid to localization, we have added a photograph of the pyramidal layer of hippocampal CA1 field labelled with alpha-bungarotoxin (which binds to alpha7 neuronal nicotinic Ach receptors) and SAF-cAMP, now included as the new Supplementary Figure 3, added a comment in the text (lines149-150) and methods (lines 372-373). Hence, the number of the subsequent figures has been modified accordingly. However, the focus of this manuscript is not on the single cells, but on the differential pattern of distribution among the different brain areas. Please see above for the discussion on the use of multiple fluorescent labelling. Concerning subcellular localization, the only technique which can spatially resolve cell membrane and pre- or post-synaptic localization of protein is electron microscopy, which was out of focus in the present work, since it requires a specific harsh treatment of the tissue. In our previous works, we were unable to obtain significant penetration in the brain tissue of the antibodies used here, and were able to perform electron microscopy only in cell lines (see Fig. 2 in Mucignat-Caretta et al 2008, Neuro-Oncology 10, 958–967). We are actively exploring different other antibody sources with the aim of performing these experiments in the future, even if we are not so confident that working on adult brain will result in feasible double labelling for electron microscopy. We tried it so many times in different experienced laboratories at four Universities over the years, that we only hope we’ll be able to do it in the future.
REV. 2: 4) Figure 6 is interesting but requires control experiments using 8-Br-cAMP.
AUTHORS: We retrieved the photographs and data of a control experiment with 8-Br-cAMP and added 6 panels (4 new images and 2 graphs) in the new supplementary 6 image (panels I to N) and a comment in the main text (lines 225-228).
REV. 2: Minor comment
The authors seemed to use several types of fluorescent cAMPs (SAR-cAMP, SAF-cAMP and 8-Fluo-cAMP). The authors should indicate which probe was used in each experiment.
AUTHORS: We apologize for our oversight. Now this information has been included wherever lacking: in the text (figure legends, main text and supplementary) we have added the relevant indications at lines: 87-88, 110-115, 168-169, 181-182, 226, 234-235.
Round 2
Reviewer 2 Report
I have no further concern.